# Properties of the Ignimbrites in the Architecture of the Historical Center of Arequipa, Peru

**Rosa Bustamante** [1,*], **Patricia Vazquez** [2] and **Nicanor Prendes** [3]

1 Escuela Técnica Superior de Arquitectura, Universidad Politécnica de Madrid, Av. Juan de Herrera, 28040 Madrid, Spain

2 GEGENAA EA3795, Université de Reims Champagne-Ardenne 2, Esplanade Roland Garros, 51100 Reims, France; patricia.vazquez@univ-reims.fr

3 Ministerio Para la Transición Ecológica (MITERD-OECC), Avda San Juan de La Cruz, 10, 28071 Madrid, Spain; nprendes@telefonica.net

* Correspondence: rosa.bustamante@upm.es

**Abstract:** The petrographic and petrophysical characteristics of three varieties of ignimbrites used in the architectural heritage of Arequipa (southwest Peru) are analyzed. The modal classification QAFP and TAS diagram discriminate their dacitic nature. Mercury injection porometry revealed very high porosity: 46.5% for white and beige ignimbrites, and 35.5% for the pink variety. Ignimbrites contain intrusions of vulcanodetrital fragments and vacuoles that influence their predominantly non-linear mechanical behavior. The results of water absorption by capillarity (C) and ultrasound pulse velocity (UPV) demonstrate a slight anisotropy for the beige variety and near isotropy for white and pink ignimbrites, which justify the randomness of the application of the ashlars in the masonry and in the selection of the faces to carve. Surfaces with hollows in the white and beige ignimbrites are the result of the erosion of the acicular pumice that fills the vacuoles.

**Keywords:** ignimbrite; petrography; physico-mechanical properties; color; cultural heritage





## 1. Introduction

Arequipa ignimbrite began to be used in vaulted constructions in the 17th century [1], although the city was founded in the middle of the 16th century during the Spanish viceroyalty. The city is surrounded by three volcanoes—Misti, Chachani and the extinct Pichu Pichu—in the central volcanic zone of the Andes in southwestern Peru. The abundance of white ignimbrite called "sillar de Arequipa" [2] made it possible to cut ashlars, voussoirs and pavement tiles measuring $40 \times 40 \times 20$ cm$^3$, which were dimensions based on the Spanish yard (vara de Castilla, 0.835905 m).

The characteristics of the carving of the church and stately house façades that represent the Hispanic American Baroque have been extensively studied, with its architecture identified by "very white ashlar stone" [3] (Figure 1a,b). Genetically, the ignimbrites are associated with volcanic trends of variable petrological series in terms of their composition [4–6], which affects their petrophysical and mechanical properties. These properties are in turn conditioned by their degree of consolidation [7], which determines their suitability as construction materials [8–10]. Currently, areas of the now-dormant quarry fronts are open to tourists.

The city is located in a seismically active zone, and earthquakes have caused significant damage for centuries; one of the oldest was recorded in 1471 [11], and the most recent of greater intensity in 2001. The last major reconstruction of the main monuments in the historical center lasted two decades after earthquakes in 1958 and 1960.

In the second half of the 19th century, with the arrival of the railway to the city, ignimbrite was used on flat roofs with ashlars between cast iron joists. Starting in the 1870s, the use of face-ashlar became popular, due to the attractive white color, without the

pigmented lime paint traditionally used in cochineal red and indigo blue colors, especially in residential buildings.

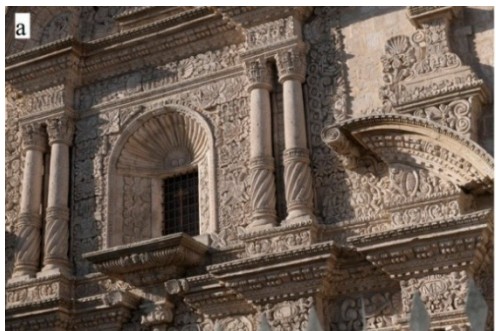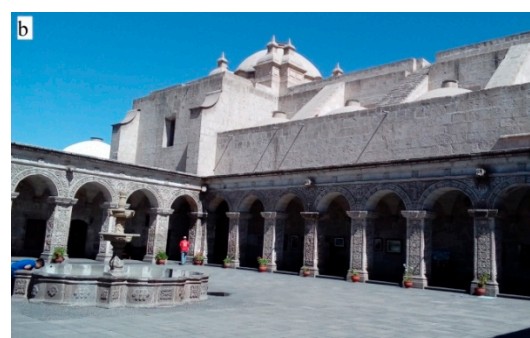
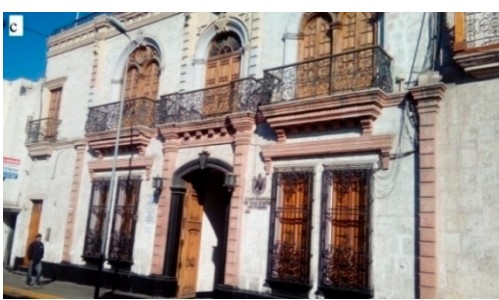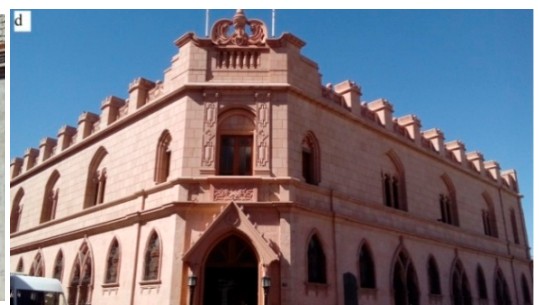

**Figure 1.** (**a**) Carved white ignimbrite façade of La Compañía de Jesús church, ca. 1698; (**b**) restored cloister after 1960 earthquake; (**c**) pink ignimbrite pilasters, corbels and moldings; (**d**) neogothic Bishop's Palace (the latter two from the first quarter of the 20th century).

This article compares the differences of the white, beige and pink ignimbrites (Figure 1c,d) that were used in the historical center (declared a World Heritage Site by UNESCO in 2000), which require ongoing building conservation and restoration works. It can contribute to aging studies [8] and the application of treatments to eliminate deposits due to environmental contamination [12] on the surfaces of the pink and beige varieties, which have been little studied.

There are several ignimbrite quarries, but the largest is in the Añashuayco stream and is also called AAI (Arequipa Airport Ignimbrite); others are RCI (Río Chili Ignimbrite), YT (Yura Tuffs) and LJI (La Joya Ignimbrite) [6] (Figure 2a). The Añashuayco quarry to the south of the Chachani volcano is one of eleven streams with volcanic deposits [13] in the surroundings of the potentially active volcano with an altitude of 6057 masl, which is made up of several stratovolcanoes. AAI is composed of two units—white ignimbrite and pink ignimbrite—covered by some alluvial deposits on top (Figure 2b). The quarry location is set between UTM coordinates 8,192,365 North and 223,806 East at 2545 masl on the slope of the volcano and UTM 8,182,237 North and 214,495 East at 1969 masl [14], along 18 km at the SW direction down to the Uchumayo zone.

Geomorphologically, they are planar beds or alluvial deposits that overlap each other, which means that, topographically, the oldest series in age are found higher up the volcano (white ignimbrite), close to the mouth or vent, and are overlapped by the younger series (pink ignimbrite). The average age of the white unit using fission track dating is $2.41 \pm 0.08$ Ma [2] and $1.65 \pm 0.04$ Ma (on biotite) for pink ones from the isotope ratios $^{40}$Ar–$^{39}$Ar [6].

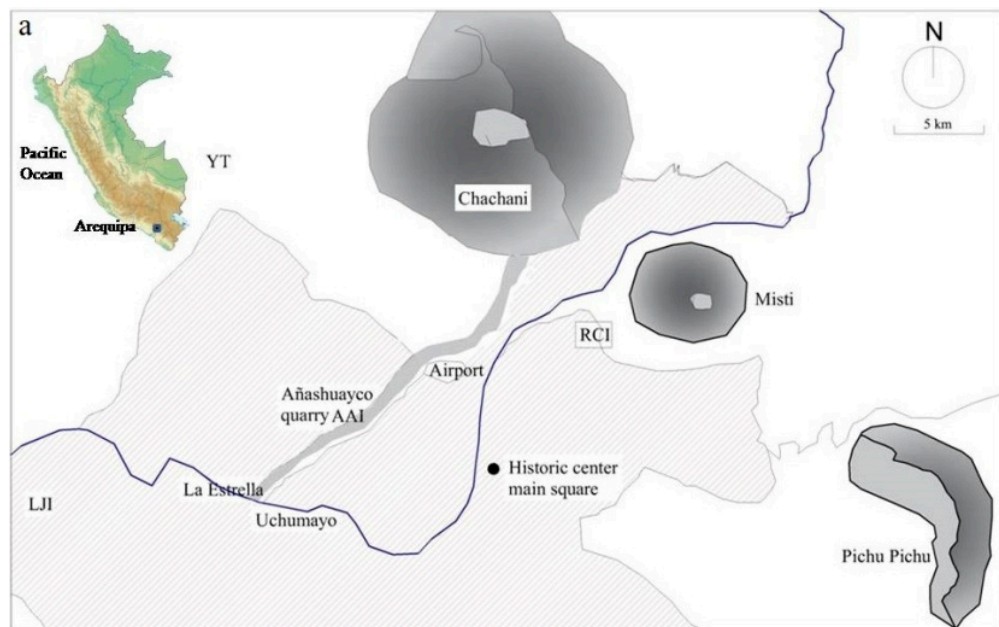

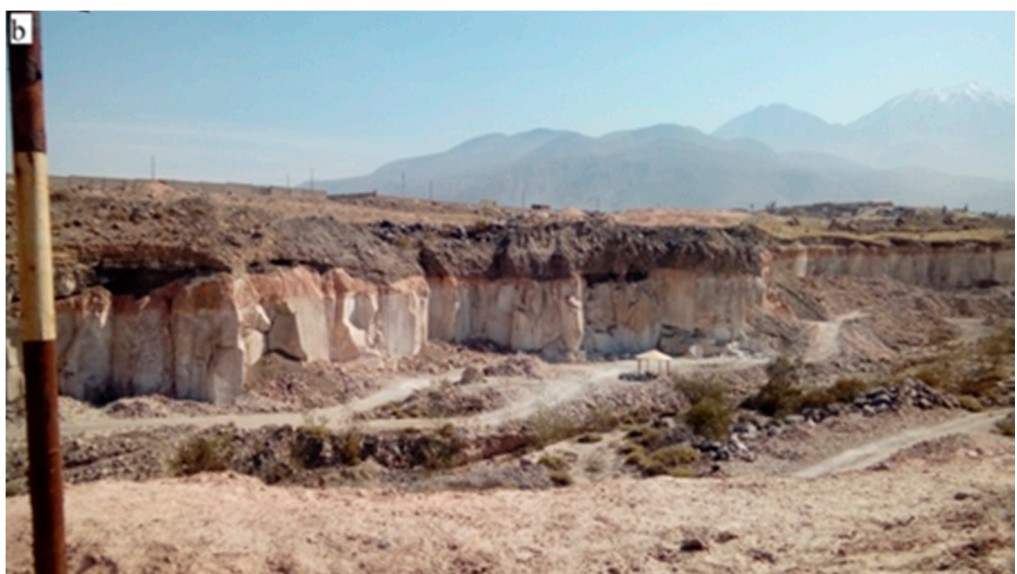

**Figure 2.** (**a**) Location. (**b**) Western front of Añashuayco quarry (Chachani volcano to the right).

## 2. Materials and Methods

Eight prismatic specimens measuring $40 \times 40 \times 160$ mm$^3$ and 24 cubic specimens with a 50 mm edge were prepared from the samples of white ignimbrite from the Añashuayco quarry (Wt samples) and beige ignimbrite from the La Estrella quarry in the area furthest from the volcano (Be samples). Only six cubic samples of pink ignimbrite (Pk samples of Uchumayo) were used, due to limited availability (Figure 2a).

Petrographic characterization through optical polarized microscopy (OM) was performed using a Carl Zeiss Jena AMPLIVAL Pol-U. For the morphological determination, scanning electronic microscopy (SEM) was required, using JEOL JSM 5400 and a magnification range from 15X to 200,000X in both backscattered electron (BSE) and secondary electron (SSE) mode with a ZEISS DSM 94 supported by EDX Oxford ISIS-Link; X-ray diffraction (XRD) with powder diffractometer Bruker D8Advance used the DIFFRACplus software.

The preliminary chemical quantification required portable X-ray fluorescence equipment: Thermo Scientific Niton XL3t 980 with a geometrically optimized large area drift detector (GOLDD+) energy-dispersive XRF (EDXRF). The analyzer works up to 50 kV

and 40 μA. The calculation of elemental concentrations was made from the "Test All Geo" integrated algorithm specific for stone materials.

In addition, the $Na^{\pm}$, $K^{\pm}$, $Ca^{2\pm}$ and $Fe^{2+}$ concentrations were measured by inductively coupled plasma atomic emission spectrometry (ICP-MS, Iris Advantage, Thermo Fisher Scientific). The analysis was conducted through three measurements of each solution.

The porous network distribution [15] was assessed, using Mercury Porosimetry (Micromeritics Autopore IV 9500), with detection ranges (at 414 MPa) between 0.005 μm and 360 μm, i.e., those pore accesses that are smaller than 360 μm.

Apparent density, water absorption at atmospheric pressure and the water absorption coefficient by capillarity measurements were determined following standards UNE-EN 1936, UNE-EN 13755 and UNE-EN 1925, respectively. Mechanical characterization, such as flexural strength under concentrated load and uniaxial compressive strength, were estimated following standards UNE-EN 12372 and UNE-EN 1926, respectively. The ultrasound pulse velocity (m/s) was measured using Pundit Plus with frequency transductors of 54 MHz. The anisotropy index (AD) for capillary water uptake and ultrasound velocity was calculated following Benavente et al. [16] based on the ratio between the minimum and maximum values for each sample.

Surface hardness was measured using a Leeb tester KH-100, which is a dynamic method relating the rebound (Rp) and the impact velocity (lp) of 75 g (type D impact device) with a result on the Leeb scale ($HL_D = 1000$ Rp/lp $\times$ 1000). Considering Leeb rebound hardness (LRH), it is more accurate, compared to Shore hardness tests [17].

The color was quantified using a Minolta CR-200 (D65 illuminant, with a diffuse beam of 8 mm in diameter, a viewing angle geometry θ [18]. It was measured using the CIE L* a* b* system. The parameter L*, or luminosity, records the values from 0 (black) to 100 (white); a* and b* are the color indices given in Cartesian coordinates (a* ranging from −60 (green) to +60 (red) and b * ranging from −60 (blue) to +60 (yellow), respectively). Given the textural heterogeneity, 100 measurements were taken for each stone type. The total color variation (ΔE*), particularly significant between white and beige ignimbrites, was calculated as follows:

$$\Delta E^* = [(\Delta L^*)^2 + (\Delta a^*)^2 + (\Delta b^*)^2]^{\frac{1}{2}} \tag{1}$$

The parameter ΔE* indicates the total color variation with a threshold of perception of the color difference for the human eye with values greater than ΔE* = 3 [19].

## 3. Results

### 3.1. Macroscopic Description

All ignimbrites have heterometric stone fragments, from a few mm to approximately 50 mm, with pronounced angular edges, although some are also rounded (Figure 3a,b), which stand out, due to their black or brown color. White and beige ignimbrites differ from pink ignimbrites, due to the presence of vacuoles (Figure 3c) of varying dimensions that come from the accumulation of gases in lava expulsion processes and that later give rise to hollows filled with pumice with secondary products of mineralization in the form of spherulitic crystals that form acicular growths of different colors (yellow, white or brown), as seen in Figure 3c and in the two samples of white ignimbrite in Figure 3d.

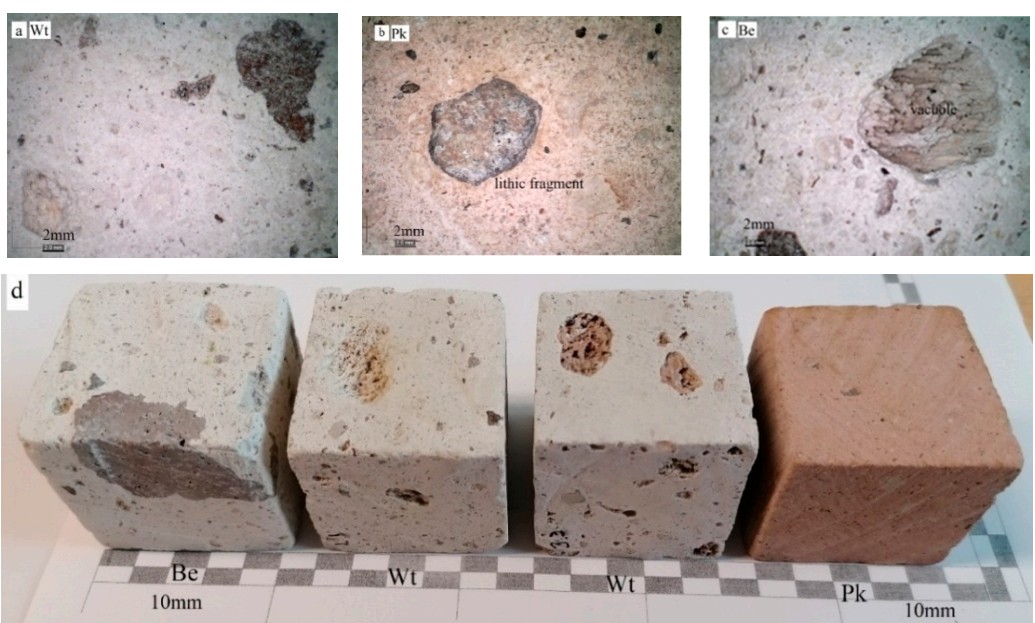

**Figure 3.** (**a**,**b**) Intrusions; (**c**) vacuole with acicular pumice; (**d**) surface appearance of Be ignimbrite with edge intrusion, two Wt ignimbrites with filled vacuoles of different colored fibbers and Pk ignimbrite without vacuoles (from left to right).

### 3.2. Petrographic Description

Ignimbrite is compact, with minerals (phenocrysts) integrated in a siliceous hypocrystallineaphanitic matrix and recrystallizations and reactive belts (Figure 4a) that develop secondary mineralogical phases filling the porous network. Microscopy (OM and BSE-SEM) confirms the geochemistry and vitreous character of the siliceous matrix (cristobalite and tridymite) as high temperature quartz phases (Q), as well as the presence of alkaline feldspars (sanidine) and plagioclase (albite-oligoclase type), philosilicates (biotites and muscovites) and opaque minerals (hematites).

Petrographically, it is not possible to differentiate the white variety from the beige variety since both have "re-assimilated" andesite fragments, as illustrated in the images (Figure 4b,c). This is unlike pink ignimbrite, which develops volcanic glass aggregates and relic glass phases without apparent deformation, pointing to a slow and homogeneous cooling consolidation process (Figure 4d). The porosity, type of mineralogical phases and their degree of consolidation with the siliceous matrix, observable in the SEM images (Figure 4e,f), suggest that the differences in behavior are more associated with post-depositional processes than with a genetic origin.

X-ray diffraction confirms the mineralogy and geochemistry of the ignimbrites, defining the variety of the type of quartz (cristobalite) and feldspar phases. Semi-quantitatively, on the diffractogram, the white and beige ignimbrites have identical peak heights and reticulate and projection spacing. This sequence of reflections supports the dominance of the albite phase in pink ignimbrite. Further analysis of acicular "pumice" crystallizations within the vacuoles reveal amphibolite mafic minerals and sodalite phases (Table 1).

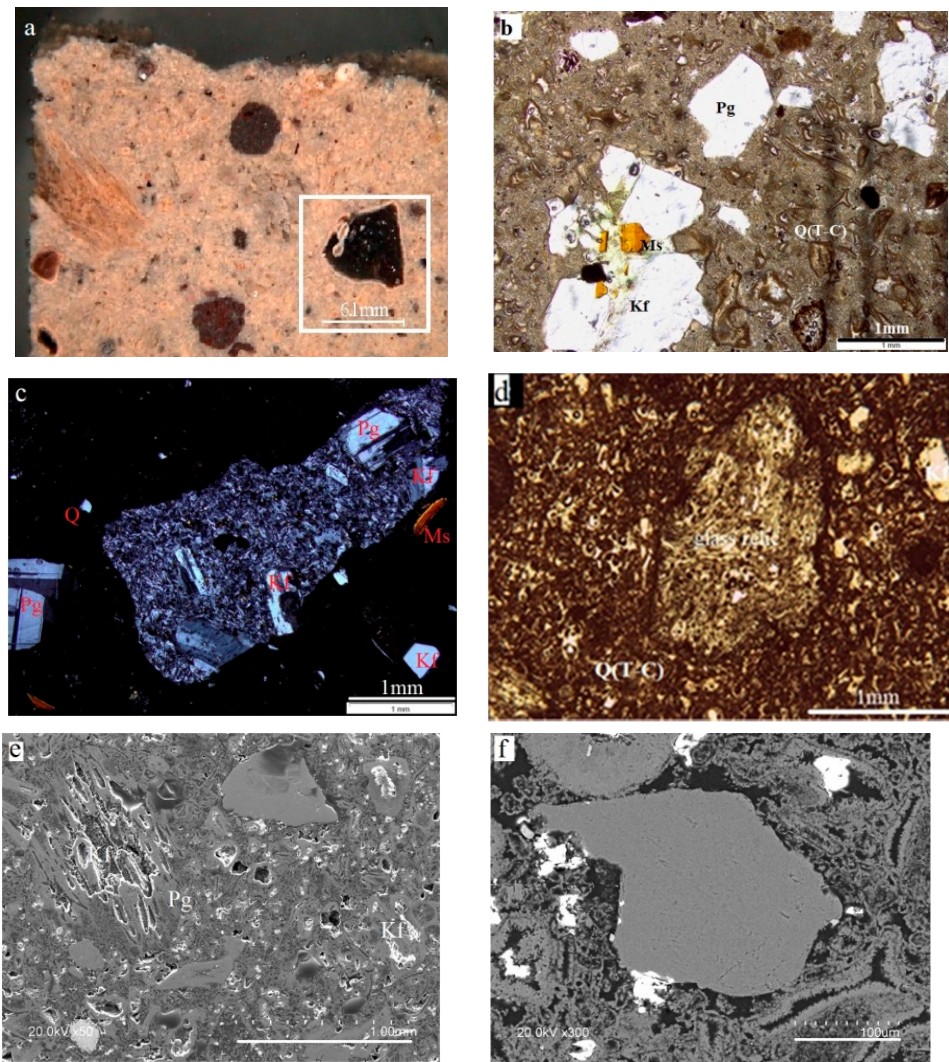

**Figure 4.** (**a**) Reactive belt; (**b**) OM-PPL of Wt ignimbrite; (**c**) XPL Be ignimbrite; (**d**) Pk ignimbrite, PPL of glass relic; (**e**) SEM/SSE of microporous network; (**f**) SEM/BSE of morphology of pores and phases (Q (T–C)) quarzt–trydimite–cristobalite, (Pg) plagioclase, (Kf) potassic feldspar, (Ms) muscovite.

**Table 1.** Results of XRD semi-quantitative analysis.

| Ignimbrite Sample Minerals (%) | Wt | Be | Pk | Vacuole Pumice |
|---|---|---|---|---|
| Cristobalite $\alpha$ SiO$_2$ | 15 | 15 | 5 | 16 |
| Tridymite SiO$_2$ | 2 | 2 | 7 | |
| Potassic Feldspar (Sanidine) K(AlSi$_3$)O$_8$ | 22 | 21 | 14 | 26 |
| Plagioclase (Albitehigh) Na(AlSi$_3$O$_8$) | 51 | 54 | 66 | 47 |
| Phyllosilicates K (Mg, Fe)$_3$(AlSi$_3$O$_{10}$)(OH)$_2$ Biotite and Muscovite | 8 | 7 | 8 | 6 |
| Hematite Fe$_2$O$_3$ | 2 | 1 | | |
| Amphiboles Ca$_2$(Mg, Fe, Al)$_5$(Al, Si)$_8$O$_{22}$(OH)$_2$ | | | | 4 |
| SodaliteNa$_8$Al$_6$Si$_6$O$_{24}$Cl$_2$ | | | | 1 |

### *3.3. Classification of Ignimbrites*

The petrographic classification was performed using steorology [20] (Figure 5) by means of thin sections (polished surfaces), which yields the modal mineralogical composition (representative mean of the sample values of each sheet studied). The representation thereof, in the QAPF diagram, assigns these samples to very similar composition and

characteristics, corresponding to dacite rocks. Variations in Q content are less than 10%, and the alkali feldspar/plagioclase ratio is 5%, allowing for the logical local composition deviations associated with petrography.

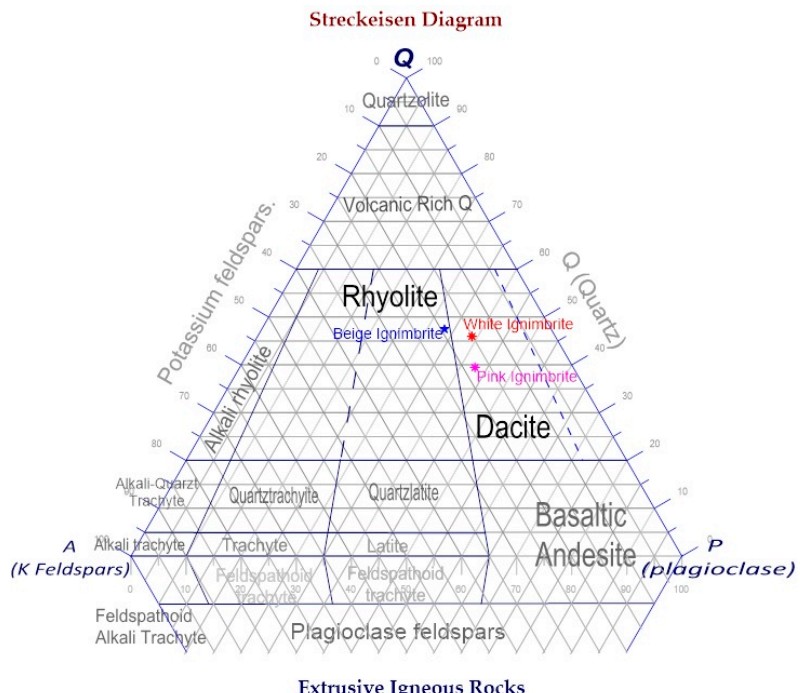

**Figure 5.** Classification of ignimbrites and their identifying field based on the percentages of mineralogical phases found (ternary diagram QAPF).

The chemical analysis of these same samples using spectroscopy, p-XRF and ICP-MS, is shown in Table 2. This made it possible to establish the rocks' classification following the total alkali–silica diagram (TAS) developed by Le Maître et al. for volcanic rocks [21] (Figure 6). The classification was deduced from the constituent oxides of the alkaline cations versus the silica content (silicon oxide or $SiO_2$). The results, consistent with the petrographic classification, revealed a dacitic composition with lower concentrations of $K_2O$ and $Na_2O$ than other ignimbrites characterized, such as that of Hınıs [22].

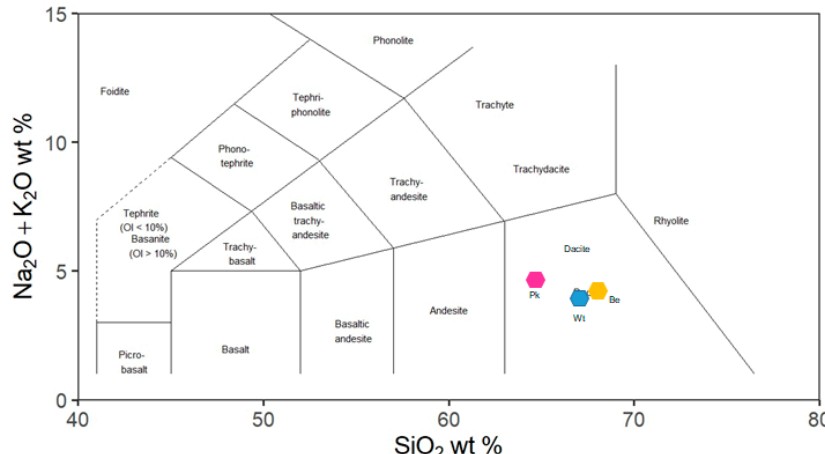

**Figure 6.** TAS classification of the three ignimbrites.

**Table 2.** Results from chemical analysis for the main elements.

| Ignimbrites | SiO$_2$ | K$_2$O | Na$_2$O | Fe$_3$O$_4$ |
|---|---|---|---|---|
| Wt | 66.9% [a] | 3.6% [a] 3.3% [b] | 0.4% [b] | 0.9% [a] 1.0% [b] |
| Be | 67.9% [a] | 3.8% [a] 3.7% [b] | 0.4% [b] | 1.0% [a] 1.1% [b] |
| Pk | 64.4% [a] | 3.7% [a] 2.8% [b] | 0.8% [b] | 0.8% [a] 0.8% [b] |

a: Results from p-XRF; b: results from ICP-MS.

### 3.4. Mercury Porometry

The mercury injection porosimetry revealed very high porosities for the three varieties, with values around 46.5% for the white and beige and 35.5% for the pink variety. The pore size distribution is shown in Figure 7. The three rocks exhibited a bimodal distribution, with similarities for the white and beige specimens.

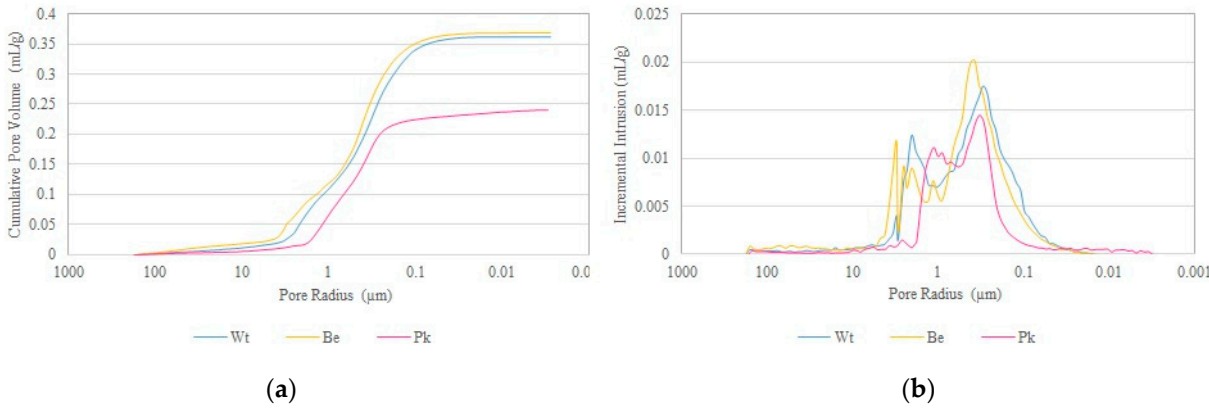

(**a**)                                                    (**b**)

**Figure 7.** Hg porosimetry curves for the three ignimbrites. (**a**) Cumulative function. (**b**) Normal function associated with the pore range.

The Wt ignimbrite presented a main connected porosity to mercury in the range of 3.72–0.03 µm with 95% of the pore throat sizes. Within these values, two families can be described. The first one corresponded to mesopores from 3.72 µm to 1 µm with a modal peak at 2.02 µm and 25% of the total voids, and the second one corresponded to micropores from 1 µm to 0.02 µm with a modal peak at 0.33 µm and 70% of the connected porosity.

The Be ignimbrite showed a main range of porosity between 6.5 µm and 0.022 µm, slightly wider than the Wt ignimbrite, and with 94% of the total measured porosity. The bimodal distribution described an irregular mesopore family from 6.5 µm to 0.9 µm with a modal peak at 3 µm that corresponded to 28% of the connected porosity, and a micropore family from 0.9 µm to 0.022 µm with a peak at 0.4 µm that made up the remaining 66%.

The Pk ignimbrite had narrower pore radii access than the other ignimbrites (Figure 7), with a range of measured porosity concentrated between 1.9 µm and 0.1 µm. The bimodal distribution was focused on a mesopore group from 1.9 µm to 0.57 µm with a peak at 1.12 µm that made up 40% of the pore volume, and the micropore group from 0.57 µm to 0.1 µm with a peak at 0.32 µm and 46% of the pore volume. Thus, this Pk ignimbrite showed greater homogeneous porosity equally distributed between meso and micropores in relation to the Wt and Be ignimbrites.

### 3.5. Physico-Mechanical Properties

The apparent density showed values between 1310 kg/m$^3$ and 1350 kg/m$^3$ for beige and white ignimbrites, respectively, and 1518 kg/m$^3$ for pink ignimbrites: values lower than those of the Morelia ignimbrites [23].

Capillary water uptake revealed slight differences between the three samples (Figure 8). Thus, for 60 min of testing, Wt ignimbrite showed a continuous and straight-line relation between weight increase and the time. Be ignimbrite exhibited the beginning of the stabi-

lization of the water uptake at about 45 min, which indicated the filling of trapped porosity; in addition, this ignimbrite showed the highest capillary coefficient, indicating higher kinetics than the other two varieties of ignimbrites, which can influence the formation of this trapped porosity. With a capillary coefficient similar to the Wt ignimbrite, the Pk variety exhibited the same water uptake kinetics, although stabilization arrived after approximately 30 min of testing.

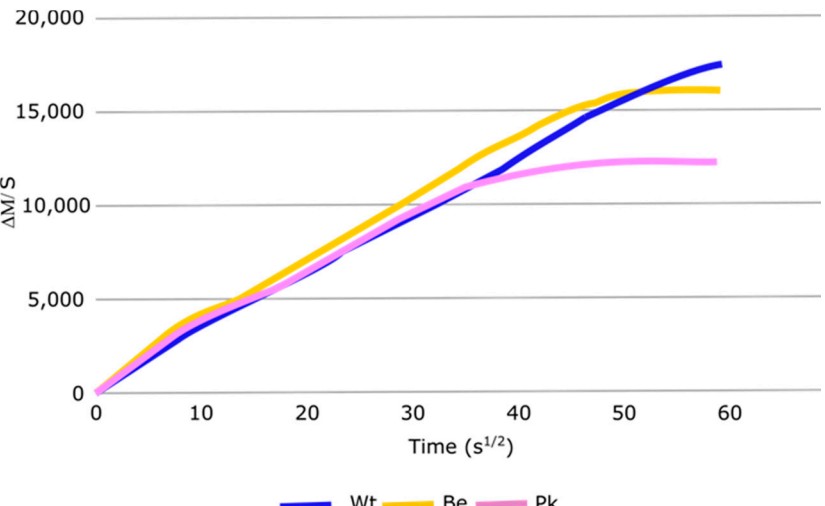

**Figure 8.** Time-dependent capillary water.

The water absorption by capillary rise (C), after 30 min of testing revealed that the Pk variety reached nearly the same saturation as in free immersion after 1 h, with values of capillary absorption around 15%. The Wt and Be varieties presented a capillary absorption of 18% and 21%, respectively, at 30 min, still far from the free immersion values (Ab) and in agreement with the kinetics shown in Figure 8.

The ultrasound pulse velocity (UPV) was related to capillary suction (C), with higher values for lower capillarity kinetics. Thus, Be ignimbrite showed the lowest values, followed by Wt and lastly Pk, which that had the highest velocity (Table 3). UPV measurements are practically similar in the three spatial directions of the samples. The threshold that marked the separation between the ignimbrites is >2000 m/s for the Pk and <2000 m/s for the Wt and Be ignimbrites.

**Table 3.** Comparison of results of Arequipa ignimbrites characterization.

| Variety | $\rho$ kg/m$^3$ | $A_b$ (%) | C g/m$^2 \cdot$s$^{0.5}$ | UPV m/s | $F_s$ MPa | UCS MPa | $E_{dyn}$ GPa |
|---|---|---|---|---|---|---|---|
| Wt [7] | 1200 1300 | | | | | | |
| Wt [8] | 1240 | 25.23 | | | | 2.43 | 9.34 |
| Wt [9] | 1260 | 27.88 | | | | 1.38 | 8.79 |
| Wt [10] | 1260 | 31 ± 0.26 | 429 ±5 | | | 2.14 | 9.35 |
| This study | | | | | | | |
| Wt | 1350 | 25 ± 1.46 | 272 ± 13 | 1853 ± 29 | 1.5 ± 0.5 | 7 ± 2 | 4.63 |
| Be | 1310 | 27 ± 2.38 | 295 ± 17 | 1709 ± 53 | 2 ± 0.5 | 5 ± 1 | 3.64 |
| Pk | 1518 | 16 ± 0.49 | 275 ± 1 | 2141 ± 34 | | 13 ± 3 | 7 |

Apparent density ($\rho$); absorption water immersion ($A_b$); capillary suction (C); ultrasound pulse velocity (UPV); flexural strength (Fs); uniaxial compressive strength (UCS); dynamic modulus of elasticity ($E_{dyn}$).

The anisotropy index (AD) expresses the behavior of this variable for the Pk and Wt varieties for UPV and capillary coefficient (C) with values higher than AD > 0.9, while Be

samples showed slightly anisotropic behavior, with AD values up to 0.85 for UPV and 0.87 for C.

The breakage of the prismatic specimens of Be ignimbrite in the flexural test is inclined at approximately 30°, conditioned by the presence of glassy grains and large vacuoles, and its strength was slightly higher than the Wt ignimbrite. Results of 1.5 MPa to 2MPa for Wt and Be ignimbrites were higher than those obtained for the Bitlis ignimbrite [24] and volcanic tuffs [25].

Permeability can be estimated theoretically using capillary data (*C*). Cueto et al. [26] proposed an accurate correlation for volcanic tuffs with characteristics similar to the studied ignimbrites, as indicated below:

$$\sqrt{k} = C \times \left(6 \times 10^{-6}\right)(m/s) \tag{2}$$

The values obtained are 2.66, 3.13 and 2.72 ($10^{-6}$ m/s) for the white, beige and pink ignimbrites, respectively.

In the uniaxial compressive strength tests, the lowest values were found for the Be ignimbrites and the higher for Pk ignimbrites (Table 3). The fracture is both granular (glassy grains) and intergranular depending on the hardness of the component minerals. In volcanic rocks, non-linear behavior has been observed [27,28], as seen in the stress-strain curves in these three cases. First, the elastic limit is more clearly seen in the Pk ignimbrite, which corresponds to brittle fracture (Figure 9a), with a defined yield point coinciding with the ultimate strength and then a plastic phase before fracture. Second is a small elastic phase and then a plastic phase that culminates in the ultimate strength before the fracture (Figure 9b). Third, there is an irregular elastic deformation as an addition of deformations in the elastic phase (Figure 9c).

Applying the formula E = 2.88Vp-1.92 [29] and knowing the ultrasound pulse velocity of the samples, the results have a standard deviation of 0.5–0.6 for the Wt and Pk ignimbrites and 0.9 for the Be ignimbrites, compared to the $E_{dyn}$ obtained in this study (Table 3).

Following purely theoretical criteria and based on their mineralogical nature, the most compact materials produce a higher rebound velocity than porous materials. The average hardness of the ignimbrites is $LH_D = 329 \pm 36$. The combination of Leebs surface hardness and UPV defines the high or low cohesion of ignimbrite components and their surface hardness, respectively, more than isolated measurements (Figure 10). Additionally, there is good relationship between UCS and LRH [30,31]. If we apply the correlation UCS = $7 \times 10^{-7} HLD^{2.8751}$ [16], knowing the results of $HL_D$ of ignimbrites, 16 MPa is obtained for the Pk ignimbrite, coinciding with the result of this study and an increase of 3 MPa more for the Wt and Be ignimbrites.

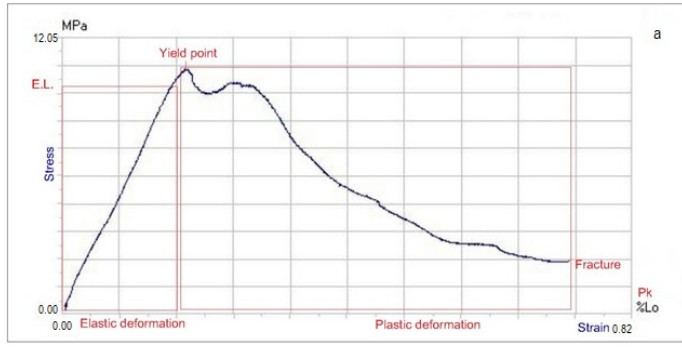

**Figure 9.** *Cont.*

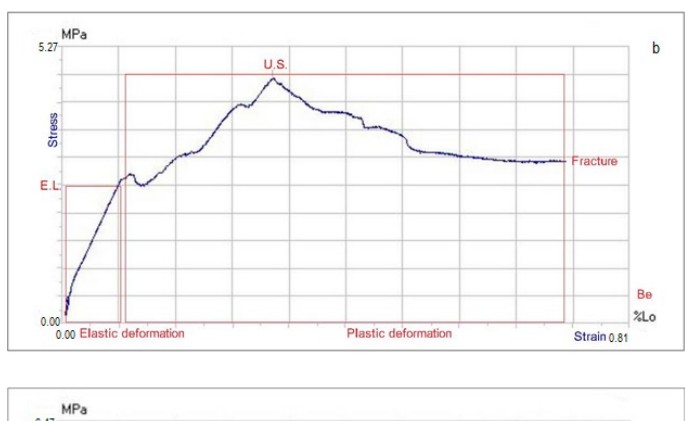

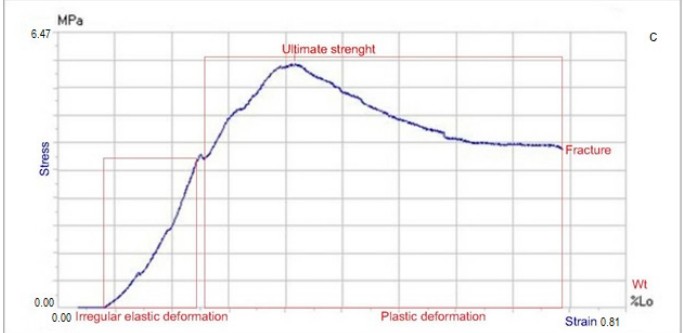

**Figure 9.** Stress–strain curves. (**a**) Pk; (**b**) Be and (**c**) Wt ignimbrites. E.L.: elastic limit.

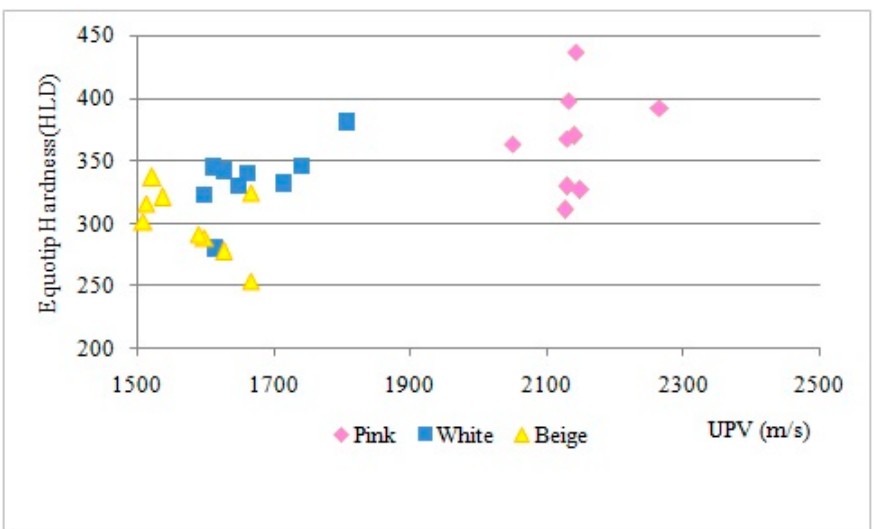

**Figure 10.** UPV vs. HLD.

*3.6. Color Measurement*

The measurements in dry conditions showed light materials with pale colors that vary for each. The white variety had the highest values of lightness with L* > 80, with low values of a*, indicating that no tendency to red or green was observed and slight positive values of b*, implying a weak yellowish appearance. For the beige ignimbrite, the values were slightly lower for lightness, and a noticeable b* value was measured in agreement with its yellowish appearance. The pink ignimbrite showed the darkest values and high enough chromatic parameters, according to the pink-red color of the stone. The difference between the specimens can be observed with the naked eye, and the results of ΔE* are in agreement with this fact (Table 4).

**Table 4.** Results of color measurement.

|  |  | L* | a* | b* |
|---|---|---|---|---|
| Wt | Avrg. | 80.2 | 1.3 | 4.8 |
|  | st.dev. | ±2.1 | ±0.2 | ±0.9 |
| Be | Avrg. | 78.4 | 1.8 | 9.3 |
|  | st.dev. | ±1.6 | ±0.2 | ±0.8 |
| Pk | Avrg. | 66.3 | 7.4 | 11.8 |
|  | st.dev. | ±0.9 | ±0.4 | ±0.5 |
|  | Wt-Be | Be-Pk | Wt-Pk |  |
| ΔE* | 4.95 | 13.51 | 16.76 |  |

## 4. Discussion

Compositionally, there is no significant variability between the white and beige samples. The content of quartz and alkaline feldspars (sanidine) and plagioclase (albite) is similar in both in semi-quantitative terms. The quartz phase, about 15% of cristobalite, has a proportion of less than 5% in pink ignimbrites, within the typical tolerance range of these rocks.

The pink ignimbrite deposits on the quarry top have a higher density than the Wt and Be ignimbrites. This result is associated with consolidation and compaction processes, which lead to sintering and compression of the pyroclastic material (due to thermal processes of transition to the glassy phase) by quenching [32], which are conditioned by the crystallochemical composition of the glass phase [6]. The mudflows descending the slopes of the volcano would have contributed to compacting the ignimbrite Pk. To this, we must also add the degassing process, which is local and depends on the overlying pyroclastic sediment column; accordingly, beige ignimbrites are characterized by containing more vacuoles. Associated with diagenetic processes, post-sedimentary sheet weathering supports the cationic leaching of unstable phases and directional hardening due to chemical cementation and modifies the porous network, resistance and durability [24,33].

The ignimbrites are classified as type II, the characteristics of which establish a density of 1.25–1.65 kg/cm$^3$, porosity of 0.34–0.50 and UCS of 1.8–9.8 MPa [32]. A double process occurs in the non-linear mechanical behavior of ignimbrites, against the compression load. While vacuoles facilitate the spread of cracks, intrusions of fragments of vulcanodetrital contribute to dissipating energy in the manner of ball joints, opposing the continuity of cracks, so the material maintains its cohesion and even reaches an ultimate resistance exceeding the elastic limit until intergranular breakage occurs (Figure 11a).

If the correlation of low dry density values, UCS comprised between 1 MPa and 5.5 MPa, higher porosity than 50% and ultrasonic velocities lower than 2000 m/s define the ignimbrites as weak rocks [34], then the ignimbrite Be is characterized in this field, but not the Pk ignimbrites.

Although the mechanical resistance of ignimbrites is low, in colonial architecture the masonry is mixed, composed of two leaves, each 20 cm thick with a core filled with pebbles and broken ashlars bonded with coarse mortar (Figure 11b). Therefore, the walls with thickness of 1.50 m to 1.75 m counteract as a whole the thrusts of the semicircular arches and the barrel vaults.

The results of the capillary suction (C) and the ultrasonic pulse velocity (UPV) that revealed a slight anisotropy for the Be variety and near isotropy for the Wt and Pk ignimbrites make it possible to assert the lack of directional flows in the quarry. This guarantees the random laying of the ashlars in the masonry and the voussoirs in the vaults, including the carved faces, as a preferential orientation to withstand compression loads was not found.

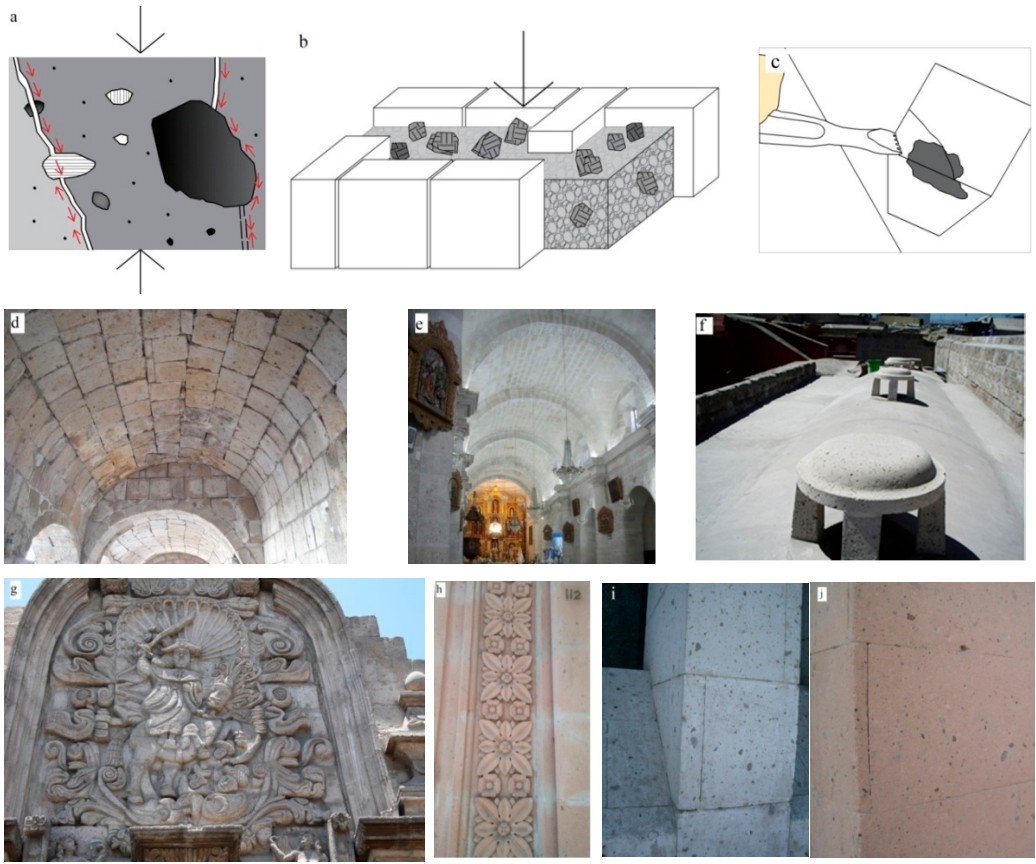

**Figure 11.** (**a**) Intergranular and through vacuole breakage; (**b**) mixed masonry; (**c**) carved with sharp claw chisel; (**d**) barrel vault; (**e**) intrados with transverse ribs; (**f**) ignimbrite skylight on the vault; (**g**) colonial carving of the James the Apostle (La Compañía de Jesús); (**h**) carved quadrifolia from the first half of the 20th century; (**i**,**j**) Wt and Pk ignimbrite cladding.

The low hardness of ignimbrites facilitates carving since the use of hammers, sharp chisels and claw chisels is enough to cut and smooth the lithic intrusions without causing the detachment of the hard grains due to the good adhesion in the matrix (Figure 11c).

All the properties previously described provide great lightness to the ignimbrite, which influenced the massive use of vaulted constructions in religious, stately and residential buildings (Figure 11d). The option of using face-ashlar is justified indoors by the white color that contributes to increasing luminosity, especially in religious naves (Figure 11e). Although the vaults do not have roofing, the thick layer of lime–pozzolan mortar on the extrados protects the voussoirs from rainwater (Figure 11f).

The colonial façades were carved by assembling the carved faces of the ashlars, blocks in the shape of T and L or in polyhedra of different dimensions with relief depth between 2 cm and 8.5 cm [35] (Figure 11g); this technique was replicated on the neocolonial façades (Figure 11h). The cohesion of the ignimbrites of high porosity is demonstrated in the mechanical sawing of the slabs of different thickness and dimension in façade cladding (Figure 11i,j).

Despite the high absorption coefficient to water by immersion, the reference parameter used in the choice of materials to demonstrate their suitability for outdoor use (close to 30% for Wt and Be ignimbrites), and high capillarity suction, the use of these ignimbrites is justified by the scarcity of rainfall in this area, registering an annual average of 102 mm [36] (Figure 12) from 2001 to 2018 (only surpassed in 2012 with 303.20 mm), an average temperature of 16.1 °C (2004–2018) with the highest temperature of 17.3 °C recorded in 2016, coinciding with the El Niño climate phenomenon, accompanied by an average relative humidity of 50% characteristic of a Andean dry climate. To avoid capillary suction, non-porous stones are used in the wall base and plints. In addition to this theoretical analysis, it

is deduced that pink ignimbrite (with recrystallization, growth and filling of the micropores in the lower void fractions) is more suitable in areas susceptible to the development of dampness and alterations associated with capillary processes of the soil or foundations.

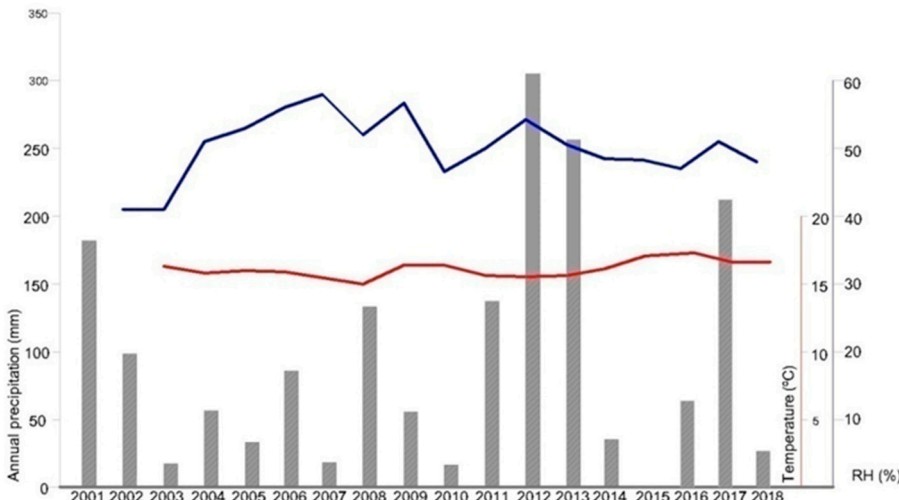

**Figure 12.** Annual precipitation, temperature and relative humidity data.

Another interesting factor from the point of view of the habitability of buildings is that the use of these ignimbrites contributes to thermal insulation since the Wt ignimbrite has a value of thermal conductivity of $0.27 \pm 0.045$ W/mK [37], even lower than that of the Uşak ignimbrite [38].

The variety of color is inherent in their formation and composition, which has been documented in other geographical contexts [31,39]. Color is related to the presence (or absence) of minerals, mainly those associated with movable cations, such as iron ($Fe^{2+}$, $Fe^{3+}$), which, due to exudation or washing processes, stain the ignimbrite in the red hue in their oxidized forms [40]. In the chemical analyses, a higher content of iron was revealed in the Wt and Be varieties associated with the hematite phase, which is why these in themselves did not exhibit a reddish color. However, in the Pk variety, the iron content is in the vitrified phase, which, in this case, is the determining factor of that tonality. The presence of Q (cristobalite) gives whitish granules, not excessively consolidated or cemented, and due to their crystallochemical conditions, they fix cations of the metallic type, staining easily; this can be seen in the darkening of the white ignimbrite by the runoff of rainwater.

Although porosity (voids, cavities and fractures) most influences the physical and mechanical properties of altered volcanic rocks [41], the wear of the white and beige ignimbrites is associated with the loss of acicular phases (pumice fibers) from the vacuoles that produce the eroded appearance with hollows, which is a different process from alveolization, pitting or flaking in other stones [42]. Since ignimbrites by their nature cannot be polished, the roughness of the surfaces favors the sanding that occurs slowly in the environmental conditions of Arequipa.

From the point of view of alteration and durability, the "ignimbritic" ashlars are more stable and resistant to Arequipa's climatic and environmental conditions. The greater predominance of alkali metal cations ($Na^{\pm}$) than alkaline earth cation metals ($Ca^{\pm}$) hinders their mobility and dissolution, i.e., under these conditions, plagioclase is more stable and consequently, it disintegrates less.

Arequipa ignimbrite is representative of the Andean zone and shares the mining-petrographic and physical–mechanical properties obtained in some ignimbrites and other pyroclastic rocks [38].

## 5. Conclusions

In the ignimbrite varieties studied, differences in the petrological and physico-mechanical properties were detected, which enables them to be selected based on the destination of the ashlar in the building. In general terms, the results obtained are in the range obtained for other ignimbrites located in different geographical locations.

Statistically, the white and beige ignimbrites have similar physical, hydric and mechanical properties, although the beige showed a slight anisotropy that should be taken into account if constructions are exposed to risky conditions. Directional properties, such as capillary water uptake and ultrasound velocity revealed a higher capillary coefficient and lower ultrasound pulse velocity for the beige variety in relation to the other two. The more compact pink ignimbrite has higher values in terms of density, compressive strength and dynamic modulus of elasticity than white and beige ignimbrites. The intrusions of rock fragments justify the behavior of the plastic phase to dissipate the stress, while the vacuoles decrease its mechanical resistance to these same effects.

The use of ignimbrite in the reconstruction, rehabilitation and restoration of works is a factor of authenticity in the characterization of Arequipa's architectural cultural heritage.

**Author Contributions:** Investigation, writing, review and editing, R.B., P.V. and N.P. All authors have read and agreed to the published version of the manuscript.

**Funding:** The APC was funded by Universidad Politécnica de Madrid, project OTT P2103020037.

**Institutional Review Board Statement:** Not applicable.

**Informed Consent Statement:** Not applicable.

**Acknowledgments:** Thanks to the staff of the Materials Laboratories of the Schools of Architecture and Building Engineering at Universidad Politécnica de Madrid. Thanks also to Roberto García (Department of Geology at the Museo Nacional de Ciencias Naturales-CSIC in Spain) and architect Kelly Llerena (Arequipa city hall) for their collaboration.

**Conflicts of Interest:** The authors declare no conflict of interest.

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
