# Peer review of "Properties of the Ignimbrites in the Architecture of the Historical Center of Arequipa, Peru"

_applsci, doi:10.3390/app112210571_

Round 1

Reviewer 1 Report

Dear authors

  • At the end of the introduction part, the work done in the study is specified with a paragraph. Please add 1-2 sentences to the end of this section about revealing the novelty of this study. This will make this study more valuable. If there are studies related to this in the region, the differences can be written.
  • The conclusion part can be extended to comparisons with the information in the literature.
  • References should be written according to the journal format.
  • the reading quality of these figures should be increased.
  • With this stone, if other material properties exist in the literature, there will be significant benefits from taking them and adding them to the article.

Reviewer 2 Report

Interesting article, unusual geological object and amazing ancient architecture!

  1. The article reflects the results of the study of white and beige ignimbrites, which discriminate their dacitic nature. The combination of petrographic and petrophysical studies gives good conclusions about the structural features of the rocks. It seems that these data are of interest for both geologists and civil engineers.

  2. Taking into account the anisotropy of the properties of the studied rocks, in addition to the porosity values, data on permeability could be cited. Perhaps it would be worthwhile to describe in more detail the secondary (postsedimentary) processes in the rocks.

    But even in the absence of these data, the manuscript is quite valid.

  3. The conclusions are consistent with the evidence and arguments presented and they address the main question posed.

  4. The references are appropriate.
